# Performance Analysis of the Spanish Men’s Top and Second Professional Football Division Teams during Eight Consecutive Seasons

**DOI:** 10.3390/s23229115

**Published:** 2023-11-11

**Authors:** Ibai Errekagorri, Roberto López del Campo, Ricardo Resta, Julen Castellano

**Affiliations:** 1Society, Sports and Physical Exercise Research Group (GIKAFIT), Department of Physical Education and Sport, Faculty of Education and Sport, University of the Basque Country (UPV/EHU), Lasarte 71, 01007 Vitoria-Gasteiz, Spain; 2Department of Competitions and Mediacocach, LaLiga, Torrelaguna 60, 28043 Madrid, Spain; rlopez@laliga.es (R.L.d.C.); rresta@laliga.es (R.R.)

**Keywords:** team sport, match analysis, collective behaviour, evolution, mixed model

## Abstract

The present study aimed to analyse the performance of the Spanish men’s top (*LaLiga1*) and second (*LaLiga2*) professional football division teams for eight consecutive seasons (from 2011–2012 to 2018–2019). The variables recorded were Passes, Successful Passes, Crosses, Shots, Goals, Corners, Fouls, Width, Length, Height, distance from the goalkeeper to the nearest defender (GkDef) and total distance covered (TD). The main results were that (1) *LaLiga1* teams showed lower values of Length from 2013–2014, and lower values of GkDef and TD from 2014–2015; (2) *LaLiga2* teams showed fewer Passes and lower values of GkDef and TD from 2014–2015, and fewer Goals and lower values of Length from 2015–2016; and (3) *LaLiga1* teams showed more Passes, Successful Passes, Shots and Goals and higher values of TD compared to *LaLiga2* teams during the eight-season period. This study concludes that *LaLiga1* teams showed fewer final offensive actions, *LaLiga2* teams showed fewer Passes and Goals and the teams of both leagues played in a space with greater density (meters by player), covering less distance as the seasons passed. The information provided in this study makes it possible to have reference values that have characterised the performance of the teams.

## 1. Introduction

The era of technology has allowed sports such as football to carry out more precise and objective studies about the performance of players and teams during competition [1]. Tracking systems technology (global navigation satellite systems or global positioning systems, local positioning systems and semi-automatic video cameras) has been allowed in sports in general and in football in particular, based on the recorded positioning data, either in geographic coordinates (latitude and longitude) or Cartesian (x and y axes), the analysis of kinematic variables (e.g., displacements, accelerations), as well as individual (e.g., heat maps) and collective (e.g., average positioning of the players) tactical variables of a team (distances between players and/or spaces covered by a group of players) [2,3,4]. The use of performance indicators obtained from this technology is essential to evaluate the performance of players and teams in competition [5], and even to carry out longitudinal monitoring.

Previous studies have explored the development of the game of football throughout the years [6,7,8,9,10,11,12,13,14]. The physical performance has received close attention regarding this longitudinal viewpoint [6,7,8,9,11,13]. Some authors have studied the physical performance evolution of the English *Premier League* teams throughout seven seasons [7], bearing in mind the specific position of players [9], or considering how teams ended up at the end of the season [8]. Barnes et al. [7] reported that the distance covered by the teams in the English *Premier League* had not changed much throughout the seven years (from 2006–2007 to 2012–2013), this way increasing the number of high-intensity actions and accumulated distance, as well as the number of sprints and accumulated distance. Moreover, the accumulated distance at high intensity and the number of high-intensity and sprint actions significantly increased in all player positions in the English *Premier League* teams throughout this period [9]. Bradley et al. [8], for their part, found that all the English Premier League teams increased the accumulated distance at high intensity when they did not have the ball. Nevertheless, those teams that finished fifth through eighth at the end of the season showed a slight increase in short distance covered at high intensity in ball possession compared to other teams, and a significant increase in accumulated sprint distance compared to other teams. Regarding the Spanish *LaLiga1* (men’s top professional football division), a recent study [11] has also analysed the physical performance evolution of the teams throughout eight seasons, in addition to the physical performance evolution of the players considering their playing position and the physical performance evolution of the teams taking the final league ranking into account. Lago-Peñas et al. [11] observed a small decrease in the total distance covered by the teams of the Spanish *LaLiga1* with a higher number of high-intensity efforts as the seasons progressed, specifically from 2012–2013 to 2019–2020. Furthermore, these authors observed an increase in the number of actions at high intensity for all positions analysed, in addition to observing a decrease in the total distance covered and an increase in the distance covered at high intensity for almost all positions. Finally, they found that the Spanish *LaLiga1* teams made a higher number of high-intensity efforts as the seasons progressed, and the Upper-Middle ranked teams (from 6th to 10th) and Lower ranked teams (from 16th to 20th) covered a greater distance at high intensity.

The technical–tactical performance has also received considerable attention in the scientific literature [7,8,9,10,11,12,14]. Some works have analysed the technical–tactical performance evolution of the teams in the English *Premier League* [7], in addition to the technical–tactical performance evolution of the teams considering the final league ranking [8], and the technical–tactical performance evolution of the players according to their position [9]. Barnes et al. [7] found that in the English *Premier League,* there was an increase in the number of passes and their effectiveness throughout seven seasons (from 2006–2007 to 2012–2013), with a notable increase in short- and medium-distance passes. However, Bush et al. [9] analysed the evolution of physical and technical performance parameters in the English *Premier League* between 2006–2007 and 2012–2013 and observed moderate–large-magnitude increases in the total number of passes performed and moderate increases in the pass success rate for central defenders and central midfielders. Bradley et al. [8] observed that during the seven seasons analysed, the first four teams classified in the English *Premier League* demonstrated the highest levels of technical performance (i.e., number of Passes and Successful Passes), but the greatest increases in the technical parameters of Passes made and received were shown by the teams classified between the fifth and eighth positions. In the Spanish *LaLiga1*, for its part, Lago-Peñas et al. [11] have also recently analysed the technical–tactical performance evolution of the teams, of the players taking their playing position into account and of the teams considering the final league ranking. These authors observed a small increase throughout the eight-season period in technical variables such as passes, long passes, passing accuracy, aerial duels and interceptions. Furthermore, they found a slight decrease during this period in technical variables such as Shots, tackles and clearances. However, Lago-Peñas et al. [11] observed that the teams classified between the first and fifteenth positions showed fewer shots, tackles and clearances, and more short passes, long passes and aerial duels as the seasons progressed. Finally, these researchers also investigated the evolution of the technical parameters considering playing positions, and they found that external midfielders and forwards significantly decreased the Shots performed in the last seasons analysed. 

To the knowledge of the authors, no investigation has analysed the evolution of the technical–tactical and physical performance of the two professional Spanish football leagues and the comparison between them over a long period. Therefore, this study aimed to analyse the Spanish men’s top (*LaLiga1*) and second (*LaLiga2*) professional football division teams’ performance considering some key competitive performance variables over a continuous period of eight seasons. 

## 2. Materials and Methods

### 2.1. Sample

For the objective of this study, all teams’ performances in the Spanish *LaLiga1* and *LaLiga2* across eight consecutive seasons (from 2011–2012 to 2018–2019) were analysed. All matches where the information required was not available were excluded, as well as matches where one or more players were sent off. As a result, out of a possible 13,472 performances (6080 in the Spanish *LaLiga1*: 20 teams, each playing 38 matches throughout the eight seasons; and 7392 in the Spanish *LaLiga2*: 22 teams, each playing 42 matches throughout the eight seasons), a total of 11,019 performances (5518 in the Spanish *LaLiga1* and 5501 in the Spanish *LaLiga2*) were analysed, representing 82% of all the possible matches. The data to carry out this study were collected for convenience. 

Data were obtained from the Spanish *Professional Football League*, which authorised the use of the variables included in this investigation. Following its ethical guidelines, this investigation does not include information that identifies football players. This study was conducted in accordance with the Declaration of Helsinki and approved by the Ethics Committee of the *University of the Basque Country* (UPV/EHU). 

### 2.2. Variables

The variables used in this work were grouped into four dimensions: Technical–Tactical (Passes, Successful Passes, Crosses and Shots), Set Piece (Goals, Corners and Fouls), Collective Tactical Behaviour (Width, Length, Height and distance from the goalkeeper to the nearest defender (GkDef)), and Physical (total distance covered (TD)). Table 1 shows the definitions of the variables for each dimension.

### 2.3. Procedures

Location and motion data were obtained by the computerised multi-camera tracking system *TRACAB* (*ChyronHego*, New York, NY, USA) and events were obtained by the data company *OPTA* (*Opta Sports*, London, UK), both using *Mediacoach* software (*LaLiga*, Madrid, Spain). The reports were generated using *Mediacoach*, for the predefined performance indicators. The reliability of the *OPTA* system has been previously proved [15] and the reliability of the multi-camera tracking system *TRACAB* has also been tested for positioning and physical performance of the players [16]. The generated reports were exported into a *Microsoft Excel* spreadsheet (*Microsoft Corporation*, Washington, DC, USA) to configure a matrix.

### 2.4. Statistical Analysis

The statistical analysis was conducted using the software *jamovi 2.4.8* [17] for *Windows*. A linear mixed model was carried out for each dependent variable in order to analyse the differences in teams’ match performance according to the league and season. League and season were considered as fixed effects and team as random effect. The Akaike information criterion (AIC) [18] and a likelihood ratio test [19] were used to select the model that best fitted each variable. The maximum likelihood (ML) estimation was used for model comparison and, for the final model of each variable, the best model again using restricted maximum likelihood (REML) estimation was refitted [19]. Marginal and conditional R^2^ metrics [20] were provided for each linear mixed model as a measure of effect sizes. Marginal R^2^ is concerned with the variance explained by fixed effects, and conditional R^2^ is concerned with the variance explained by both fixed and random effects [20]. The level of significance was set at *p* < 0.05.

## 3. Results

Table 2 shows the effects of season for each league and the effects of league on the variables of the Technical–Tactical dimension. On the one hand, the Spanish *LaLiga1* teams showed fewer Crosses in 2016–2017 (−2.720; *p* = 0.013), 2017–2018 (−2.907; *p* = 0.008) and 2018–2019 (−2.348; *p* = 0.032) compared to the 2011–2012 season, and fewer Shots in 2014–2015 (−1.327; *p* = 0.052) compared to the 2011–2012 season. On the other hand, the Spanish *LaLiga2* teams showed fewer Passes in 2014–2015 (−33.205; *p* = 0.031), 2015–2016 (−37.535; *p* = 0.015), 2016–2017 (−32.181; *p* = 0.036), 2017–2018 (−31.649; *p* = 0.038) and 2018–2019 (−30.408; *p* = 0.046) compared to the 2011–2012 season, and fewer Successful Passes in 2014–2015 (−35.293; *p* = 0.037) and 2015–2016 (−41.131; *p* = 0.015) compared to the 2011–2012 season. Likewise, the Spanish *LaLiga2* teams showed fewer Passes (−28.445; *p* < 0.001), Successful Passes (−36.810; *p* < 0.001) and Shots (−0.808; *p* < 0.001) compared to the Spanish *LaLiga1* teams during the whole period analysed. See Figure 1 to facilitate the interpretation of the Technical–Tactical variables’ results when comparing the seasons with each other and one league with the other.

Table 3 shows the effects of season for each league and the effects of league on the variables of the Set Piece dimension. On the one hand, the Spanish *LaLiga1* teams showed fewer Corners in 2016–2017 (−0.886; *p* = 0.001), 2017–2018 (−0.560; *p* = 0.042) and 2018–2019 (−0.721; *p* = 0.009) compared to the 2011–2012 season, and fewer Fouls in 2015–2016 (−1.019; *p* = 0.017), 2017–2018 (−0.859; *p* = 0.044) and 2018–2019 (−1.160; *p* = 0.007) compared to the 2011–2012 season. On the other hand, the Spanish *LaLiga2* teams showed fewer Goals in 2015–2016 (−0.210; *p* = 0.009), 2016–2017 (−0.225; *p* = 0.004), 2017–2018 (−0.181; *p* = 0.019) and 2018–2019 (−0.207; *p* = 0.008) compared to the 2011–2012 season, and fewer Corners in 2014–2015 (−0.562; *p* = 0.005) compared to the 2011–2012 season. Likewise, the Spanish *LaLiga2* teams showed fewer Goals (−0.195; *p* < 0.001) and more Fouls (0.880; *p* < 0.001) compared to the Spanish *LaLiga1* teams during the eight seasons analysed. See Figure 2 to facilitate the interpretation of the Set Piece variables’ results when comparing the seasons with each other and one league with the other.

Table 4 shows the effects of season for each league and the effects of league on the variables of the Collective Tactical Behaviour dimension. On the one hand, the Spanish *LaLiga1* teams showed lower values of Length in 2013–2014 (−0.629; *p* = 0.049), 2014–2015 (−0.895; *p* = 0.005), 2015–2016 (−1.544; *p* < 0.001), 2016–2017 (−1.919; *p* < 0.001), 2017–2018 (−1.954; *p* < 0.001) and 2018–2019 (−2.122; *p* < 0.001) compared to the 2011–2012 season, lower values of Height in 2014–2015 (−1.390; *p* = 0.011) compared to the 2011–2012 season, and lower values of GkDef in 2014–2015 (−2.339; *p* < 0.001), 2015–2016 (−2.047; *p* < 0.001), 2016–2017 (−1.790; *p* < 0.001), 2017–2018 (−1.565; *p* < 0.001) and 2018–2019 (−1.728; *p* < 0.001) compared to the 2011–2012 season. On the other hand, the Spanish *LaLiga2* teams showed lower values of Width in 2014–2015 (−1.557; *p* = 0.002), 2015–2016 (−1.557; *p* = 0.002) and 2016–2017 (−0.951; *p* = 0.054) compared to the 2011–2012 season, lower values of Length in 2015–2016 (−0.977; *p* < 0.001), 2016–2017 (−0.532; *p* = 0.051), 2017–2018 (−0.555; *p* = 0.038) and 2018–2019 (−0.915; *p* < 0.001) compared to the 2011–2012 season, lower values of Height in 2014–2015 (−1.017; *p* = 0.009), 2016–2017 (−1.070; *p* = 0.006) and 2017–2018 (−0.761; *p* = 0.047) compared to the 2011–2012 season, and lower values of GkDef in 2014–2015 (−1.655; *p* < 0.001), 2015–2016 (−1.229; *p* < 0.001), 2016–2017 (−1.392; *p* < 0.001), 2017–2018 (−1.298; *p* < 0.001) and 2018–2019 (−1.162; *p* < 0.001) compared to the 2011–2012 season. See Figure 3 to facilitate the interpretation of the Collective Tactical Behaviour variables’ results when comparing the seasons with each other and one league with the other.

Table 5 shows the effects of season for each league and the effects of league on the variable of the Physical dimension. On the one hand, the Spanish *LaLiga1* teams showed lower values of TD in 2014–2015 (−3160.020; *p* < 0.001), 2015–2016 (−2057.377; *p* = 0.011), 2016–2017 (−2273.371; *p* = 0.005), 2017–2018 (−2227.386; *p* = 0.006) and 2018–2019 (−2882.185; *p* < 0.001) compared to the 2011–2012 season. On the other hand, the Spanish *LaLiga2* teams showed lower values of TD in 2014–2015 (−3214.570; *p* < 0.001), 2015–2016 (−3387.602; *p* < 0.001), 2016–2017 (−4099.113; *p* < 0.001), 2017–2018 (−3278.626; *p* < 0.001) and 2018–2019 (−4262.594; *p* < 0.001) compared to the 2011–2012 season. Likewise, the Spanish *LaLiga2* teams showed lower values of TD (−698.705; *p* = 0.016) compared to the Spanish *LaLiga1* teams during the whole period analysed. See Figure 4 to facilitate the interpretation of the Physical variable’s results when comparing the seasons with each other and one league with the other. 

## 4. Discussion

The aim of this study was to analyse the performance of the Spanish professional teams (*LaLiga1* and *LaLiga2*) over a continuous period of eight seasons (from 2011–2012 to 2018–2019). The main results of this study were that (1) the Spanish *LaLiga1* teams showed lower values of Length from the third season (from 2013–2014 to 2018–2019), and lower values of GkDef and TD from the fourth season (from 2014–2015 to 2018–2019); (2) the Spanish *LaLiga2* teams showed fewer Passes and lower values of GkDef and TD from the fourth season (from 2014–2015 to 2018–2019), and fewer Goals and lower values of Length from the fifth season (from 2015–2016 to 2018–2019); and (3) the Spanish *LaLiga1* teams showed more Passes, Successful Passes, Shots, Goals and higher values of TD compared to *LaLiga2* teams during the eight-season period. 

Regarding the Technical–Tactical dimension, the distribution in the variables such as Passes and Successful Passes made by the teams of the Spanish *LaLiga1* represents a performance stability throughout the period analysed, coinciding with a previous study [11] performed also in the Spanish *LaLiga1* a few years earlier. Barnes et al. [7] found in the English *Premier League* a significant increase in the total and Successful Passes in the last two seasons analysed (2011–2012 and 2012–2013) compared to the first season (2006–2007), with a moderate effect size. These differences described between the English and Spanish leagues may be due to the fact that the periods were consecutive, making it likely that this progression has been slowed in the last 15 years. On the contrary, the results of the present study showed that the Spanish *LaLiga2* teams significantly decreased the total number of Passes from the 2014–2015 season. These results coincide with the greater prominence of the defensive phase of the game in the second division [21], which could be related to better performance in competition. 

In the case of Crosses, it is worth mentioning that *LaLiga1* teams performed fewer actions of this variable from the 2016–2017 season. With regard to the Shots, both leagues showed a stable trend over the eight seasons. Again, these results are similar to those of Lago-Peñas et al. [11] and Barnes et al. [7]. Therefore, it is worth noting that the trend of the Shots in these two works was also quite stable throughout the period studied. Considering the difficulty of describing the way teams play in competition based on a few variables [22], it seems that as the seasons passed, the Spanish *LaLiga1* teams showed a lower degree of effective offensive play, represented by a similar number of Passes and fewer Crosses. However, it seems that the Spanish *LaLiga2* teams were characterised by a direct style of play as the seasons passed, represented by a smaller number of Passes and similar Crosses and Shots. Additionally, although the teams of *LaLiga2* decreased the number of Passes, they showed similar accuracy throughout the eight seasons (i.e., a similar number of Successful Passes). 

When comparing the Technical–Tactical variables between leagues during the period studied, the Spanish *LaLiga1* obtained significantly higher values than the Spanish *LaLiga2* in Passes, Successful Passes and Shots. These results are similar to those of Castellano and Casamichana [21]. However, it should be noted that these authors qualify that it was actually the top 10 teams in the first division that showed significantly higher values in Shots, total Passes and Successful Passes (measured as a percentage) than the other three groups. In English football, some researchers [23] found that the players in the English *Premier League* performed more total and Successful passes than the players in the English *Championship* (the second division of England’s league system) and *League 1* (the third division of England’s league system). According to different studies published previously [24,25,26], it seems that Passes, Successful Passes or Shots made, among other variables, are directly related to success or a higher competitive level of the teams. In addition, Castellano [27] found that variables such as Passes, Successful Passes or Shots had a strong relationship with the final classification in the Spanish men’s professional football division, while they did not have it in the Spanish men’s second professional football division. However, this study only analysed two seasons, thus not allowing any trend to be established. Therefore, it is worth mentioning that the Spanish *LaLiga1* teams stood out for showing high values in the variables of the Technical–Tactical dimension that are most related to success. 

With regard to the Set Piece Dimension, the Spanish *LaLiga1* teams showed fewer Corners in the last three seasons analysed (from 2016–2017 to 2018–2019). It has already been mentioned previously that caution must be used to describe the way teams play in competition considering certain variables, but this could be another indicator that *LaLiga1* teams showed a lower degree of effective offensive play as the seasons passed. Fewer Fouls made were also found in the Spanish *LaLiga1* in 2015–2016, 2017–2018 and 2018–2019 compared to the 2011–2012 season. It seems that *LaLiga1* teams showed more cautious and conservative defensive strategies as the seasons passed [28]. However, the Spanish *LaLiga2* teams showed fewer Goals from the 2015–2016 season. Although the teams of this league showed a similar number of Shots throughout the seasons, their efficacy decreased over time. This result, together with the reduction in the number of Passes accumulated per game, could be interpreted as a reduction in the quality of the teams in this category. When comparing the Set Piece variables between leagues during the eight-season period, on the one hand, the Spanish *LaLiga1* showed significantly higher values in Goals. This seems to support the idea suggested by Castellano [27] when he found that goals scored had a very strong relationship with the achievement of a higher number of points at the end of the league competition in the Spanish *LaLiga1*, in the sense that scoring goals brings teams closer to success. On the other hand, the Spanish *LaLiga2* showed significantly higher values for Fouls. Added to the lower number of goals per game, this could indicate that the defensive aspect tends to be more relevant in this league, in the sense that not conceding goals brings teams closer to success [27]. 

Regarding the Collective Tactical Behaviour dimension, similar values of Width but lower values of Length were found from the 2013–2014 season for the Spanish *LaLiga1* teams. It seems that they increased the density of the effective playing space (the same number of players in less area) as the seasons progressed. In *LaLiga2,* this was more remarkable, reducing both Width and Length as the seasons progressed. With regard to the Height variable, the Spanish *LaLiga2* teams showed lower values in 2014–2015, 2016–2017 and 2017–2018 compared to the 2011–2012 season. Therefore, it is worth mentioning that *LaLiga2* teams played closer to their own goal line during some seasons. A significant decrease in GkDef values was also found from the 2014–2015 season for both leagues. This could be explained by the fact that the goalkeepers of the teams in the Spanish professional football are required to play a greater role in the offensive phase of the game, demanding his participation in initiating or continuing an attack with the players closest to him, such as with his centre-backs [9], or that the teams have been able to adopt a more defensive style of play due to less ball possession during matches. It should be noted that no significant differences were found between the leagues in the eight seasons analysed. Contrary to these results, Castellano and Casamichana [21] observed higher values in the variable Width for the top-ranking teams compared to the bottom-ranking teams for both Spanish *LaLiga1* and *LaLiga2*.

Finally, in relation to the Physical dimension, the Spanish *LaLiga1* and *LaLiga2* teams showed lower values of TD from the 2014–2015 season. One reason for this may be the reduction in the effective playing time of the matches, which is known to have an outstanding effect on the physical performance accumulated by the players [29]. Lago-Peñas et al. [11] also observed a small decrease in the total distance covered from 2014–2015 to 2019–2020 compared to the 2012–2013 season. Another study [13] also found a decrease in the total distance covered by the teams of the Spanish *LaLiga1* in the 2018–2019 season compared to the previous three. Anyway, these authors analysed just four consecutive seasons, so the conclusions are not clear. On the contrary, Barnes et al. [7] observed that the total distance covered by the English *Premier League* teams remained stable during seven seasons (from 2006–2007 to 2012–2013). Allen et al. [6], for their part, observed small increases in the English *Premier League* in the last season analysed (2018–2019) compared to the first (2014–2015), and from season to season (i.e., 2016–2017 > 2015–2016 and 2017–2018 > 2016–2017). Nevertheless, these findings were not consistently significant, so it can be concluded that the total distance covered by the English *Premier League* players also remained stable over five seasons (from 2014–2015 to 2018–2019). Regarding the differences between leagues, Spanish *LaLiga1* showed significantly higher values than Spanish *LaLiga2* during the eight-season period, similar to that described by Pons et al. [13]. A greater number of Set Pieces (e.g., Fouls) in *LaLiga2*, with a greater importance of the defensive phase, could limit the time available to play (e.g., shorter effective playing time) and, consequently, reduce the accumulated distance [29]. Bradley et al. [23] found that the teams of the English *Championship* and *League 1* covered a greater total distance than the teams of *Premier League*. On this matter, it should be noted that each domestic league or country is characterised by having a particular demand of the game [30].

## 5. Limitations

The information provided in this study, especially due to the inclusion of a large volume of performances by the Spanish professional teams over eight seasons, makes it possible to have reference values that have characterised the performance of the teams in the dimensions and variables studied. In addition, to the authors’ knowledge, this is the first work to analyse the evolution of variables of the Technical–Tactical, Set Piece and Collective Tactical Behaviour dimensions of the Spanish *LaLiga2* (the men’s second professional football division). However, the present study is not without limitations. Firstly, possible differences motivated by changes in video-tracking cameras over the years that have been made must be considered [16]. Secondly, the averages of the variables were calculated to evaluate the performance of the teams, but it should be noted that the stability shown by these throughout the seasons must be interpreted with care since the inherent variability of the game makes the performances of the teams span a wide range. Thirdly, if ball possessions had been calculated [29], they would have helped to better interpret the differences in the play of the teams over the years. This subject, distinguishing the attack and defence phase, is suggested for future research. Fourth, the inclusion of other Technical–Tactical and Physical variables (e.g., recoveries, duels, types of passes, accumulated distance in high-speed running and sprint, number of accumulated high-speed runs and sprints...) and contextual variables such as the change of coach, the period of the season, the match venue or the rival’s level [31,32,33] could help refine possible inferences about the performance of the teams and to better explain their variability/stability over the years. Fifth and finally, it should be noted that although a sample of eight seasons was used in this study, caution must be taken when extrapolating these league results to other countries or competitions, since they represent the specific characteristics of the two main national leagues from Spain. Therefore, proposing this type of study in other leagues or countries could help to better understand the evolution of the game on a more global level. 

## 6. Conclusions

The main conclusion of this study is that the teams for both Spanish *LaLiga1* and *LaLiga2* presented some changes in different dimensions throughout the eight seasons analysed. It should be noted that the Spanish *LaLiga1* teams showed fewer final offensive actions such as Crosses and Corners throughout the years, while the Spanish *LaLiga2* teams showed fewer Passes and Goals. The teams of both leagues displayed their Collective Tactical Behaviour in a space with greater density, playing with the goalkeepers closer and closer to their defensive line and deploying less distance covered as the seasons passed. Finally, the Spanish *LaLiga1* stood out for obtaining higher values than *LaLiga2* in variables associated with success, such as Passes, Successful Passes, Shots and Goals during the whole period studied. The information provided in the present study makes it possible to have reference values that have characterised the performance of the teams. 

## Figures and Tables

**Figure 1 sensors-23-09115-f001:**
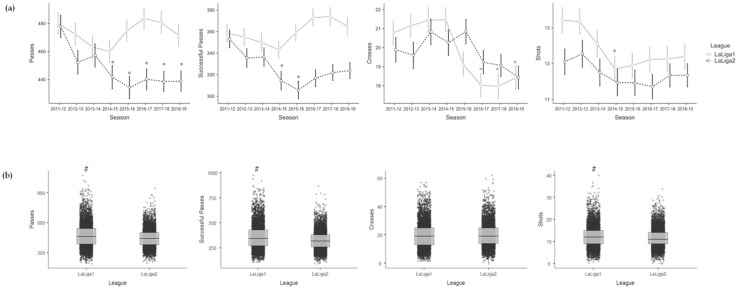
Mean and error bar (95% confidence interval) of the Technical–Tactical variables for each season and league (**a**), and boxplot with median values, interquartile ranges and outliers of the Technical–Tactical variables for each league considering the eight seasons together (**b**). * is >2011–2012 (a) and # is >*LaLiga2* (**b**) for a significance level of *p* < 0.05.

**Figure 2 sensors-23-09115-f002:**
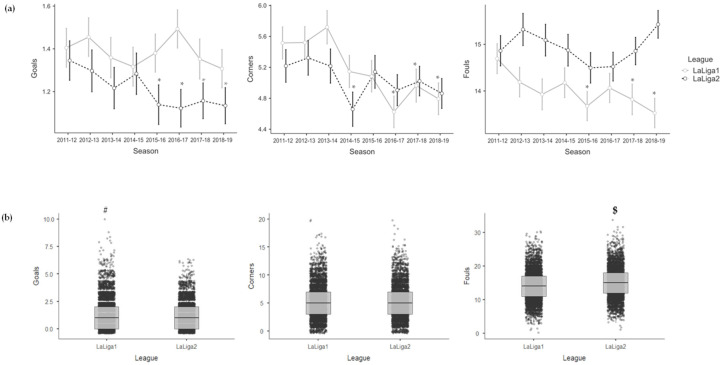
Mean and error bar (95% confidence interval) of the Set Piece variables for each season and league (**a**), and boxplot with median values, interquartile ranges and outliers of the Set Piece variables for each league considering the eight seasons together (**b**). * is >2011–2012 (**a**), # is >*LaLiga2* (**b**) and $ is >*LaLiga1* (**b**) for a significance level of *p* < 0.05.

**Figure 3 sensors-23-09115-f003:**
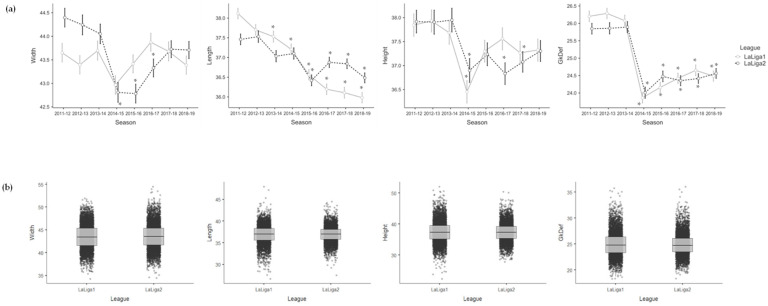
Mean and error bar (95% confidence interval) of the Collective Tactical Behaviour variables for each season and league (**a**), and boxplot with median values, interquartile ranges and outliers of the Collective Tactical Behaviour variables for each league considering the eight seasons together (**b**). * is >2011–12 (a) for a significance level of *p* < 0.05.

**Figure 4 sensors-23-09115-f004:**
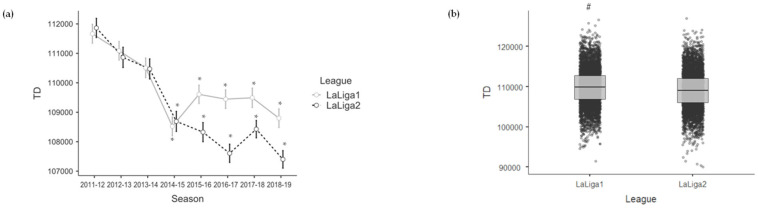
Mean and error bar (95% confidence interval) of the Physical variable for each season and league (**a**), and boxplot with median values, interquartile ranges and outliers of the Physical variable for each league considering the eight seasons together (**b**). * is >2011–2012 (**a**) and # is >*LaLiga2* (**b**) for a significance level of *p* < 0.05.

**Table 1 sensors-23-09115-t001:** Definitions of the variables for each dimension.

Dimensions	Variables	Definitions
Technical–Tactical	Passes	An intentionally played ball from one player to another with any part of the body that is allowed in the laws of the game. To calculate this variable, the total number of successful and unsuccessful passes made by the team per match is considered.
	Successful Passes	A successful pass is one that reaches its recipient. To calculate this variable, the total number of successful exchanges of the ball between two players of the same team per match is considered.
	Crosses	Any ball sent into the rival team’s penalty box from a wide position. To calculate this variable, the total number of successful and unsuccessful Crosses made by the team per match is considered.
	Shots	An attempt to score a goal is made with any part of the body that is allowed in the laws of the game, either on or off the goal. To calculate this variable, the total number of shots made by the team per match is considered.
Set Piece	Goals	Total number of goals scored by the team per match.
	Corners	A kick that is performed on a set piece from the corner of the field of play nearest to where the ball went out. To calculate this variable, the total number of corners taken by the team per match is considered.
	Fouls	Any infringement that is penalised as foul play by the referee. To calculate this variable, the total number of fouls received by the team per match is considered.
Collective Tactical Behaviour	Width	Mean team width per match, considered as the distance (in m) between the two furthest-apart players of the same team across the width of the pitch. To calculate this variable, the time in which the ball is out of play and the goalkeeper’s activity is excluded.
	Length	Mean team length per match, considered as the distance (in m) between the two furthest-apart players of the same team along the length of the pitch. To calculate this variable, the time in which the ball is out of play and the goalkeeper’s activity is excluded.
	Height	Mean team defence depth per match, considered as the distance (in m) between the furthest back player and the goal line he is defending. To calculate this variable, the time in which the ball is out of play and the goalkeeper’s activity is excluded.
	GkDef	Mean distance (in m) from the goalkeeper to the nearest defender of the same team per match. To calculate this variable, the time in which the ball is out of play is excluded.
Physical	TD	Total distance covered (in m) by all the team’s players that participated in the match, including the goalkeeper’s activity.

**Table 2 sensors-23-09115-t002:** Effects of season for each league and effects of league on the variables of the Technical–Tactical dimension.

		Passes			Successful Passes			Crosses			Shots		
*LaLiga1*	**Fixed Effects**	**Estimate**	**SE**	** *p* **	**Estimate**	**SE**	** *p* **	**Estimate**	**SE**	** *p* **	**Estimate**	**SE**	** *p* **
	Intercept	473.602	6.612	<0.001	359.966	7.122	<0.001	19.795	0.272	<0.001	12.392	0.169	<0.001
	2012–2013 - 2011–2012	−8.417	26.439	0.751	−4.479	28.481	0.875	0.396	1.085	0.716	−0.010	0.677	0.988
	2013–2014 - 2011–2012	−16.667	26.480	0.530	−8.931	28.519	0.755	0.649	1.093	0.553	−0.686	0.680	0.315
	2014–2015 - 2011–2012	−19.928	26.441	0.452	−15.075	28.483	0.597	0.681	1.086	0.532	−1.327	0.677	0.052
	2015–2016 - 2011–2012	−4.502	26.433	0.865	0.930	28.476	0.974	−1.608	1.084	0.140	−1.228	0.676	0.071
	2016–2017 - 2011–2012	3.168	26.435	0.905	13.961	28.477	0.625	−2.720	1.085	0.013	−1.092	0.676	0.108
	2017–2018 - 2011–2012	4.818	26.456	0.856	19.915	28.497	0.486	−2.907	1.089	0.008	−1.040	0.679	0.127
	2018–2019 - 2011–2012	−8.205	26.437	0.757	6.801	28.479	0.812	−2.348	1.085	0.032	−1.004	0.676	0.140
	**Random Effects**	**SD**	**Variance**	**ICC**	**SD**	**Variance**	**ICC**	**SD**	**Variance**	**ICC**	**SD**	**Variance**	**ICC**
	Team	82.297	6772.867	0.474	88.898	7902.914	0.522	3.130	9.800	0.125	1.994	3.977	0.160
	Residual	86.745	7524.762		85.054	7234.107		8.289	68.712		4.569	20.880	
	Marginal R^2^/Conditional R^2^	0.005/0.467			0.008/0.526			0.027/0.148			0.010/0.168		
*LaLiga2*	**Fixed Effects**	**Estimate**	**SE**	** *p* **	**Estimate**	**SE**	** *p* **	**Estimate**	**SE**	** *p* **	**Estimate**	**SE**	** *p* **
	Intercept	445.255	3.805	<0.001	323.227	4.197	<0.001	19.631	0.215	<0.001	11.605	0.113	<0.001
	2012–2013 - 2011–2012	−21.909	15.344	0.155	−13.436	16.907	0.428	−0.173	0.876	0.843	0.161	0.459	0.726
	2013–2014 - 2011–2012	−16.308	15.256	0.287	−12.058	16.819	0.474	0.925	0.869	0.289	−0.312	0.455	0.494
	2014–2015 - 2011–2012	−33.205	15.253	0.031	−35.293	16.816	0.037	0.373	0.868	0.668	−0.619	0.455	0.176
	2015–2016 - 2011–2012	−37.535	15.225	0.015	−41.131	16.791	0.015	1.095	0.864	0.207	−0.658	0.452	0.148
	2016–2017 - 2011–2012	−32.181	15.185	0.036	−30.524	16.754	0.070	−0.456	0.857	0.596	−0.728	0.448	0.107
	2017–2018 - 2011–2012	−31.649	15.142	0.038	−22.756	16.716	0.175	−0.569	0.850	0.504	−0.268	0.445	0.547
	2018–2019 - 2011–2012	−30.408	15.152	0.046	−19.636	16.725	0.242	−1.213	0.852	0.156	−0.326	0.446	0.465
	**Random Effects**	**SD**	**Variance**	**ICC**	**SD**	**Variance**	**ICC**	**SD**	**Variance**	**ICC**	**SD**	**Variance**	**ICC**
	Team	48.274	2330.387	0.279	53.688	2882.391	0.325	2.457	6.036	0.090	1.267	1.605	0.081
	Residual	77.583	6019.187		77.372	5986.445		7.817	61.098		4.273	18.256	
	Marginal R^2^/Conditional R^2^	0.015/0.290			0.017/0.336			0.008/0.097			0.004/0.085		
All seasons	**Fixed Effects**	**Estimate**	**SE**	** *p* **	**Estimate**	**SE**	** *p* **	**Estimate**	**SE**	** *p* **	**Estimate**	**SE**	** *p* **
	Intercept	459.380	3.705	<0.001	341.562	4.024	<0.001	19.696	0.180	<0.001	11.988	0.101	<0.001
	*LaLiga2* - *LaLiga1*	−28.445	7.411	<0.001	−36.810	8.048	<0.001	−0.189	0.361	0.600	−0.808	0.202	<0.001
	**Random Effects**	**SD**	**Variance**	**ICC**	**SD**	**Variance**	**ICC**	**SD**	**Variance**	**ICC**	**SD**	**Variance**	**ICC**
	Team	66.182	4380.008	0.393	72.182	5210.253	0.441	2.972	8.833	0.120	1.676	2.808	0.126
	Residual	82.298	6772.938		81.308	6611.072		8.056	64.900		4.423	19.564	
	Marginal R^2^/Conditional R^2^	0.018/0.404			0.028/0.456			0.000/0.120			0.007/0.132		

Notes: SE is Standard Error; SD is Standard Deviation; ICC is Intraclass Correlation Coefficient. Statistical significance set at *p* < 0.05.

**Table 3 sensors-23-09115-t003:** Effects of season for each league and effects of league on the variables of the Set Piece dimension.

		Goals			Corners			Fouls		
*LaLiga1*	**Fixed Effects**	**Estimate**	**SE**	** *p* **	**Estimate**	**SE**	** *p* **	**Estimate**	**SE**	** *p* **
	Intercept	1.383	0.046	<0.001	5.168	0.068	<0.001	14.016	0.106	<0.001
	2012–2013 - 2011–2012	0.043	0.182	0.812	0.017	0.271	0.950	−0.495	0.422	0.242
	2013–2014 - 2011–2012	−0.046	0.183	0.801	0.177	0.274	0.519	−0.736	0.426	0.086
	2014–2015 - 2011–2012	−0.086	0.182	0.638	−0.368	0.271	0.177	−0.514	0.422	0.225
	2015–2016 - 2011–2012	−0.020	0.182	0.914	−0.424	0.270	0.119	−1.019	0.421	0.017
	2016–2017 - 2011–2012	0.085	0.182	0.640	−0.886	0.270	0.001	−0.632	0.421	0.135
	2017–2018 - 2011–2012	−0.034	0.183	0.852	−0.560	0.272	0.042	−0.859	0.424	0.044
	2018–2019 - 2011–2012	−0.095	0.182	0.602	−0.721	0.271	0.009	−1.160	0.421	0.007
	**Random Effects**	**SD**	**Variance**	**ICC**	**SD**	**Variance**	**ICC**	**SD**	**Variance**	**ICC**
	Team	0.538	0.290	0.166	0.718	0.516	0.064	1.135	1.287	0.070
	Residual	1.209	1.461		2.745	7.534		4.128	17.037	
	Marginal R^2^/Conditional R^2^	0.002/0.167			0.015/0.078			0.006/0.076		
*LaLiga2*	**Fixed Effects**	**Estimate**	**SE**	** *p* **	**Estimate**	**SE**	** *p* **	**Estimate**	**SE**	** *p* **
	Intercept	1.203	0.020	<0.001	5.021	0.049	<0.001	14.900	0.095	<0.001
	2012–2013 - 2011–2012	−0.060	0.081	0.459	0.090	0.200	0.654	0.516	0.386	0.183
	2013–2014 - 2011–2012	−0.135	0.080	0.095	−0.015	0.198	0.941	0.227	0.383	0.553
	2014–2015 - 2011–2012	−0.061	0.080	0.444	−0.562	0.198	0.005	0.080	0.383	0.836
	2015–2016 - 2011–2012	−0.210	0.079	0.009	−0.070	0.196	0.723	−0.343	0.380	0.367
	2016–2017 - 2011–2012	−0.225	0.078	0.004	−0.306	0.193	0.114	−0.338	0.375	0.370
	2017–2018 - 2011–2012	−0.181	0.077	0.019	−0.166	0.189	0.383	0.051	0.371	0.890
	2018–2019 - 2011–2012	−0.207	0.077	0.008	−0.338	0.190	0.078	0.637	0.372	0.089
	**Random Effects**	**SD**	**Variance**	**ICC**	**SD**	**Variance**	**ICC**	**SD**	**Variance**	**ICC**
	Team	0.167	0.028	0.023	0.413	0.171	0.023	0.986	0.972	0.052
	Residual	1.097	1.203		2.711	7.347		4.191	17.565	
	Marginal R^2^/Conditional R^2^	0.005/0.028			0.005/0.028			0.006/0.058		
All seasons	**Fixed Effects**	**Estimate**	**SE**	** *p* **	**Estimate**	**SE**	** *p* **	**Estimate**	**SE**	** *p* **
	Intercept	1.286	0.024	<0.001	5.084	0.044	<0.001	14.455	0.072	<0.001
	*LaLiga2* - *LaLiga1*	−0.195	0.048	<0.001	−0.164	0.088	0.063	0.880	0.143	<0.001
	**Random Effects**	**SD**	**Variance**	**ICC**	**SD**	**Variance**	**ICC**	**SD**	**Variance**	**ICC**
	Team	0.393	0.155	0.104	0.641	0.411	0.052	1.082	1.171	0.063
	Residual	1.154	1.332		2.727	7.435		4.159	17.300	
	Marginal R^2^/Conditional R^2^	0.006/0.110			0.001/0.053			0.010/0.073		

Notes: SE is Standard Error; SD is Standard Deviation; ICC is Intraclass Correlation Coefficient. Statistical significance set at *p* < 0.05.

**Table 4 sensors-23-09115-t004:** Effects of season for each league and effects of league on the variables of the Collective Tactical Behaviour dimension.

		Width			Length			Height			GkDef		
*LaLiga1*	**Fixed Effects**	**Estimate**	**SE**	** *p* **	**Estimate**	**SE**	** *p* **	**Estimate**	**SE**	** *p* **	**Estimate**	**SE**	** *p* **
	Intercept	43.510	0.136	<0.001	36.923	0.079	<0.001	37.424	0.135	<0.001	25.008	0.106	<0.001
	2012–2013 - 2011–2012	−0.269	0.545	0.622	−0.413	0.316	0.194	0.104	0.541	0.848	0.070	0.423	0.868
	2013–2014 - 2011–2012	0.028	0.546	0.959	−0.629	0.317	0.049	−0.114	0.543	0.833	−0.144	0.424	0.735
	2014–2015 - 2011–2012	−0.662	0.545	0.226	−0.895	0.316	0.005	−1.390	0.541	0.011	−2.339	0.423	<0.001
	2015–2016 - 2011–2012	−0.220	0.545	0.688	−1.544	0.316	<0.001	−0.526	0.540	0.332	−2.047	0.423	<0.001
	2016–2017 - 2011–2012	0.221	0.545	0.686	−1.919	0.316	<0.001	−0.285	0.540	0.598	−1.790	0.423	<0.001
	2017–2018 - 2011–2012	0.092	0.546	0.867	−1.954	0.317	<0.001	−0.569	0.542	0.295	−1.565	0.423	<0.001
	2018–2019 - 2011–2012	−0.238	0.545	0.663	−2.122	0.316	<0.001	−0.521	0.540	0.337	−1.728	0.423	<0.001
	**Random Effects**	**SD**	**Variance**	**ICC**	**SD**	**Variance**	**ICC**	**SD**	**Variance**	**ICC**	**SD**	**Variance**	**ICC**
	Team	1.689	2.852	0.404	0.960	0.921	0.251	1.633	2.666	0.231	1.305	1.702	0.366
	Residual	2.053	4.214		1.659	2.753		2.977	8.864		1.716	2.946	
	Marginal R^2^/Conditional R^2^	0.009/0.409			0.134/0.351			0.017/0.244			0.157/0.466		
*LaLiga2*	**Fixed Effects**	**Estimate**	**SE**	** *p* **	**Estimate**	**SE**	** *p* **	**Estimate**	**SE**	** *p* **	**Estimate**	**SE**	** *p* **
	Intercept	43.534	0.123	<0.001	36.922	0.067	<0.001	37.326	0.096	<0.001	24.885	0.087	<0.001
	2012–2013 - 2011–2012	−0.114	0.495	0.817	0.016	0.270	0.952	−0.037	0.387	0.923	−0.014	0.350	0.968
	2013–2014 - 2011–2012	−0.240	0.492	0.627	−0.401	0.268	0.136	−0.016	0.384	0.967	0.121	0.348	0.729
	2014–2015 - 2011–2012	−1.557	0.492	0.002	−0.391	0.268	0.146	−1.017	0.384	0.009	−1.655	0.348	<0.001
	2015–2016 - 2011–2012	−1.557	0.492	0.002	−0.977	0.267	<0.001	−0.664	0.383	0.085	−1.229	0.348	<0.001
	2016–2017 - 2011–2012	−0.951	0.491	0.054	−0.523	0.267	0.051	−1.070	0.381	0.006	−1.392	0.347	<0.001
	2017–2018 - 2011–2012	−0.459	0.489	0.349	−0.555	0.266	0.038	−0.761	0.379	0.047	−1.298	0.346	<0.001
	2018–2019 - 2011–2012	−0.484	0.490	0.324	−0.915	0.266	<0.001	−0.599	0.380	0.117	−1.162	0.347	<0.001
	**Random Effects**	**SD**	**Variance**	**ICC**	**SD**	**Variance**	**ICC**	**SD**	**Variance**	**ICC**	**SD**	**Variance**	**ICC**
	Team	1.576	2.484	0.344	0.839	0.705	0.237	1.167	1.363	0.165	1.114	1.240	0.333
	Residual	2.178	4.744		1.506	2.268		2.629	6.910		1.576	2.484	
	Marginal R^2^/Conditional R^2^	0.041/0.371			0.037/0.265			0.020/0.181			0.106/0.403		
All seasons	**Fixed Effects**	**Estimate**	**SE**	** *p* **	**Estimate**	**SE**	** *p* **	**Estimate**	**SE**	** *p* **	**Estimate**	**SE**	** *p* **
	Intercept	43.521	0.093	<0.001	36.919	0.060	<0.001	37.371	0.084	<0.001	24.944	0.080	<0.001
	*LaLiga2* - *LaLiga1*	0.023	0.186	0.902	−0.005	0.119	0.967	−0.105	0.167	0.530	−0.126	0.160	0.430
	**Random Effects**	**SD**	**Variance**	**ICC**	**SD**	**Variance**	**ICC**	**SD**	**Variance**	**ICC**	**SD**	**Variance**	**ICC**
	Team	1.659	2.753	0.381	1.053	1.109	0.306	1.444	2.085	0.209	1.435	2.060	0.431
	Residual	2.116	4.478		1.585	2.511		2.809	7.889		1.648	2.716	
	Marginal R^2^/Conditional R^2^	0.000/0.381			0.000/0.306			0.000/0.209			0.001/0.432		

Notes: SE is Standard Error; SD is Standard Deviation; ICC is Intraclass Correlation Coefficient. Statistical significance set at *p* < 0.05.

**Table 5 sensors-23-09115-t005:** Effects of season for each league and effects of league on the variable of the Physical dimension.

		TD		
*LaLiga1*	**Fixed Effects**	**Estimate**	**SE**	** *p* **
	Intercept	109,899.711	198.923	<0.001
	2012–2013 - 2011–2012	−595.480	795.136	0.455
	2013–2014 - 2011–2012	−1040.822	797.418	0.194
	2014–2015 - 2011–2012	−3160.020	795.275	<0.001
	2015–2016 - 2011–2012	−2057.377	794.793	0.011
	2016–2017 - 2011–2012	−2273.371	794.905	0.005
	2017–2018 - 2011–2012	−2227.386	796.154	0.006
	2018–2019 - 2011–2012	−2882.185	795.086	<0.001
	**Random Effects**	**SD**	**Variance**	**ICC**
	Team	2437.959	5,943,646.440	0.312
	Residual	3623.584	13,130,363.010	
	Marginal R^2^/Conditional R^2^	0.054/0.349		
*LaLiga2*	**Fixed Effects**	**Estimate**	**SE**	** *p* **
	Intercept	109,215.122	162.056	<0.001
	2012–2013 - 2011–2012	−873.847	654.515	0.184
	2013–2014 - 2011–2012	−1206.071	650.338	0.065
	2014–2015 - 2011–2012	−3214.570	650.183	<0.001
	2015–2016 - 2011–2012	−3387.602	648.641	<0.001
	2016–2017 - 2011–2012	−4099.113	646.466	<0.001
	2017–2018 - 2011–2012	−3278.626	644.275	<0.001
	2018–2019 - 2011–2012	−4262.594	644.702	<0.001
	**Random Effects**	**SD**	**Variance**	**ICC**
	Team	2028.913	4,116,489.842	0.226
	Residual	3750.620	14,067,152.080	
	Marginal R^2^/Conditional R^2^	0.110/0.311		
All seasons	**Fixed Effects**	**Estimate**	**SE**	** *p* **
	Intercept	109,549.497	143.778	<0.001
	*LaLiga2* - *LaLiga1*	−698.705	287.555	0.016
	**Random Effects**	**SD**	**Variance**	**ICC**
	Team	2546.513	6,484,726.698	0.323
	Residual	3687.301	13,596,189.014	
	Marginal R^2^/Conditional R^2^	0.006/0.327		

Notes: SE is Standard Error; SD is Standard Deviation; ICC is Intraclass Correlation Coefficient. Statistical significance set at *p* < 0.05.

## Data Availability

Data are contained within the article.

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
