# Peer review of "Performance Analysis of the Spanish Men’s Top and Second Professional Football Division Teams during Eight Consecutive Seasons"

_sensors, 2023, doi:10.3390/s23229115_

Round 1

Reviewer 1 Report

Comments and Suggestions for Authors

The interesting topic. Studies are needed to provide reference values on the subject, as is the case in question.

However, we think that the article can be improved if the discussion does not limit itself to presenting data and referencing other studies, without in most cases an attempt to explain the results obtained. Namely, the possible explanations for the differences found between League 1 and League 2, and the studies referenced.

It is not a question of speculating, but rather of trying to explain in the light of knowledge (scientific and of some episodes or specificities that have occurred in the football leagues in Spain in the years under analysis) and the limitations of the study itself.

I also consider that the limitations of the study, instead of appearing at the end of the discussion, should be constituted as a specific and more developed point. This is very important because it would help to contextualize the discussion and give specific indications of how the conclusions can be used and future studies to be developed.

Reviewer 2 Report

Comments and Suggestions for Authors

This is a very nice paper that is well-written and of large interest to football researchers and practitioners. I have the following questions:

- Why excluding matches where players are sent off? I mean, I understand that this changes a team's performance, but does this not also belong to the reality? Could it not somehow be kept?

- In your discussion about total passes, I was wondering if the non-increase could not be related to the change away from tikitaka. Especially in Spain this was big at the years prior to 2014, so I assume this is related to the present findings.

- In the presentation of results, I cannot find AIC nor likelihood ratio tests, where are they? Also, why is RMLE needed?

- I find the discussion extremely long and one loses concentration while reading it. Would it not be worth to shorten it?

- Finally, I would suggest adding some references to general sports analytics content to attract more general attention to the paper and show it belongs to this strand. Suggestions are "Analytics Methods in Sports- Using Mathematics and Statistics to Understand Data from Baseball, Football, Basketball and other sports." and "Science Meets Sport: When Statistics are More than Numbers".

Comments on the Quality of English Language

I only detected a few typos, such as "a recent study have" on line 53, "a notably increase" on line 66. They may be more, but overall it is very well written.

Reviewer 3 Report

Comments and Suggestions for Authors

Dear authors, the comments made below have no other objective than to try to improve your document if possible.

Regarding the introduction, greater depth would be appreciated as far as possible, as well as a better thread that once brings us closer to the situation leads us to the objectives of the research. This is done but very abruptly.

Regarding the method, in the sample section, the approval by an ethics committee must appear since at the end of the text its obtaining and approval is stated, it must be made explicit, it is not dispensable information as it is a research in which human beings participated.

Likewise, it is necessary to identify what type of sampling has been carried out, and it is even advisable to explain why those seasons were chosen and not the previous or subsequent ones.

In the results chapter, the layout and presentation of the document in terms of tables must be adjusted.

On the other hand, after reading the document, the following question arises that I consider appropriate to clarify in the document:

Have promotions and relegations of clubs been taken into account? Why?

Given the results obtained, it is recommended that a graphic contribution be made to facilitate their interpretation when comparing one league with the other and even seasons with each other.

On the other hand, the research described in the document provides an excellent opportunity to go deeper through analyzes that provide more information such as logistic regression analyzes (among others) that allow us to intuit where the evolution of both competitions are headed since a first-class team It must reach the figures that you have presented both to go down, to maintain itself and to find itself in different positions in the classification.

The inclusion of the following seasons is recommended as they are atypical seasons conditioned by the COVID-19 pandemic and its new regulatory adaptations. Which would significantly enrich the document.

Comments on the Quality of English Language

The writing of the work and the construction of some structures must be reviewed

Round 2

Reviewer 3 Report

Comments and Suggestions for Authors

Dear authors,

Having seen the modifications made, I invite you to include in the main text of the document the approval of the ethics committee (which they have because they reflect it in a final minor section), due to the relevance that this type of research has when dealing with of humans.

Regarding not taking into account the modifications that entailed and entail the possibility of making five substitutions, I consider their inclusion in the present work an opportunity to provide it with greater robustness in terms of results and conclusions, so I strongly encourage you to it.
